# Suppression of *SHROOM1* Improves In Vitro and In Vivo Gene Integration by Promoting Homology-Directed Repair

**DOI:** 10.3390/ijms21165821

**Published:** 2020-08-13

**Authors:** Zhihua Zhao, Hanshuo Zhang, Tuanlin Xiong, Junyi Wang, Di Yang, Dan Zhu, Juan Li, Ye Yang, Changhong Sun, Yuting Zhao, Jianzhong Jeff Xi

**Affiliations:** 1State Key Laboratory of Biomembrane and Membrane Biotechnology, Institute of Molecular Medicine, Peking University, Beijing 100871, China; zhaozhihua1831@126.com (Z.Z.); ytzhao2014@163.com (Y.Z.); 2Department of Biomedical Engineering, College of Engineering, Peking University, Beijing 100871, China; shzzhshuo@sina.com (H.Z.); wangjunyi1994@pku.edu.cn (J.W.); coeyangdi@pku.edu.cn (D.Y.); zhudanzd1991@126.com (D.Z.); jinyujin_0@163.com (J.L.); yangye8686@163.com (Y.Y.); sunchanghong1983@163.com (C.S.); 3College of Life Science, Peking University, Beijing 100871, China; xiongtuanlin@mail.tsinghua.edu.cn

**Keywords:** homologous recombination, *SHROOM1*, knock-in, gene editing, CRISPR/Cas9

## Abstract

Homologous recombination (HR) is often used to achieve targeted gene integration because of its higher precision and operability compared with microhomology-mediated end-joining (MMEJ) or non-homologous end-joining (NHEJ). It appears to be inefficient for gene integration in animal cells and embryos due to occurring only during cell division. Here we developed genome-wide high-throughput screening and a subsequently paired crRNA library screening to search for genes suppressing homology-directed repair (HDR). We found that, in the reporter system, HDR cells with knockdown of *SHROOM1* were enriched as much as 4.7-fold than those with control. Down regulating *SHROOM1* significantly promoted gene integration in human and mouse cells after cleavage by clustered regularly interspaced short palindromic repeats (CRISPR)/CRISPR-associated protein-9 nuclease (Cas9), regardless of the donor types. The knock-in efficiency of mouse embryos could also be doubled by the application of *SHROOM1* siRNA during micro-injection. The increased HDR efficiency of *SHROOM1* deletion in HEK293T cells could be counteracted by YU238259, an HDR inhibitor, but not by an NHEJ inhibitor. These results indicated that *SHROOM1* was an HDR-suppressed gene and that the *SHROOM1* knockdown strategy may be useful for a variety of applications, including gene editing to generate cell lines and animal models for studying gene function and human diseases.

## 1. Introduction

Targeted gene integration is usually achieved by a method based on homologous recombination (HR) [1,2,3]. Highly efficient integration requires a designed nuclease to generate the specific DNA double-strand break (DSB) and a repair template that harbors left and right homology arms (HAs) (100–3000 bp). Custom-designed nucleases include zinc-finger nucleases (ZFN) [4,5], transcription activator-like effector nucleases (TALEN) [2,6], the clustered regularly interspaced short palindromic repeats (CRISPR)/CRISPR-associated protein-9 nuclease (Cas9) system [3,7,8], and the CRISPR-Cpf1 (Cas12a) system [9,10]. Once a DSB is created, externally supplied DNA fragments can be introduced around the cleavage site to participate in the HR repair. Other cell repair pathways like non-homologous end-joining (NHEJ) or microhomology-mediated end-joining (MMEJ) are also used for targeted gene integration. The former seems to elevate the efficiency obviously, but it is unable to form a precise replacement and sometimes results in unforeseeable indels, making it difficult to generate endogenous and exogenous in-frame fusion genes for chimeric protein production [11,12]. The latter can manage accurate integration easily, but has low efficiency [13,14].

HR-mediated gene integration is widely used, but is still inefficient in cells and embryos because HR only happens during the late S/G2 phase [15,16]. As a result, how to enhance the efficiency has become a focus of current research. DNA donor engineering techniques such as chemical modification [17,18,19] or double-cut site-contained [20,21] have achieved greater efficiency. Approaches that push the cell to choose the HR pathway, like the application of the NHEJ inhibitor [22,23] or cell cycle key factor [21,24], increased integration to an extent. Similarly, higher expression of the HR-essential protein [25,26,27], or fusing it with a custom-designed nuclease [25,28], could improve the efficiency.

There have rarely been reports about HR-suppressed genes and their usage in targeted gene integration, therefore, in this study, we conducted for the first time a genome-wide, high-throughput screening of genes regulating HR, and subsequently carried out paired crRNA library screening to look for HR-suppressed genes. We obtained *SHROOM1* and further tested whether its knockdown could increase HDR efficiency in mammalian cells and embryos. We found that down-regulating *SHROOM1* with siRNA could elevate HDR efficiency up to 3-fold in cultured cells, and significantly improve knock-in efficiency in mouse embryos.

## 2. Results

### 2.1. Genome-Wide Screening is Conducted to Search for Genes that Modulate HDR

To identify the factors modulating HDR efficiency at the DSB site, we modified a previously described fluorescence-based system that measures *I*-*SceI*-induced HDR events to carry out a genome-wide screening [29]. This system consists of a Tet-On [30] expression of *I*-*SceI* endonuclease and a reporter that contained one inactivated green fluorescent protein (GFP) copy linked with an internal GFP fragment (Figure 1a and Appendix A). When doxycycline (Dox) was added, the reverse tetracycline transcriptional activator (rtTA) bound to the tet-responsive element (TRE) and changed the conformation to initiate the expression of *I-SceI*. Then, the inactivated *GFP* with 18 bp *I-SceI* insertion was incised and conducted an HDR event, utilizing the distal GFP fragment to restore functional GFP expression (Figure 1a and Appendix A).

Using human osteosarcoma cell, U2OS, stably expressing the above system (iDR-GFP U2OS), we examined the levels of HDR modulation induced by individually transfecting the siRNAs of 21,257 genes by a high-content analysis instrument and calculation of fold change (FC) (Figure 1b). After treating with siRNA or NC and inducing by Dox, in each field, cells with the HDR activity which exhibited green fluorescence would be counted as N. (GFP^+^ cells). While total cells were stained with 4’,6-diamidino-2-phenylindole (DAPI) and counted as N. (total cells) (Appendix A). Six adjacent fields from a same sample were collected, and final HDR-ratio would be calculated by the ratio of N. (GFP+ cells) to N. (total cells). We used relative FC of HDR-ratio to define the ability of a gene affecting HDR (Figure 1b). *BRCA2* [31,32] and *RAD51* [32,33], critical factors potentiating recombinational DNA repair, were used as positive controls, and their siRNAs notably decreased the HDR fluorescence and resulted in FC = 0.48 or FC = 0.41, respectively, compared with NC siRNA (Figure 1c). In addition, the fluorescence brightness of two randomly selected genes (*SSTR3* and *SLC36A3*) matched the reading data well, which represented high accuracy of our screening (Figure 1c). Our screening results showed that 367 siRNAs led to FC <0.5, including well-known components of DNA resection and repair (*CtIP* [34,35] and *BRCA1* [31,35]), suggesting that these genes may maintain or promote HDR (Figure 1d, Appendix A). On the other hand, there were 1347 genes where siRNAs resulted in FC >1.5. These genes may play an important role in suppressing HDR progress in cells.

### 2.2. Suppressing SHROOM1 Promotes HDR in BFP-293T Cells

In order to confirm that primary genes were suppressing HDR, we combined a paired-crRNA library targeting the 1347 genes mentioned above and 37 controls (11,918 pairs, eight pairs targeting each primary gene transcript) with the dual-cut blue fluorescent protein (BFP) reporter [36] cells for HDR to carry out a further screening (Figure 2a). Each cell in the screening pool stably expressed a copy of *Acidaminococcus* sp. BV3L6 Cpf1 [10], as well as mono-paired crRNAs targeting the promoter or splicing site of a single gene, respectively [37] (Appendix A). To ensure the reliability of the screening, we first validated the editing efficiency of the integrated Cpf1 and mono crRNA pairs in HEK293T cells (Appendix A). Genotyping and sequencing results showed that, after 28 days, the single paired crRNA copy of *UPF1* could guide Cpf1 to generate effective editing including single cutting and long fragment deletion (Appendix A). This pool was then transfected with *Streptococcus pyogenes* Cas9 and sgRNA vectors targeting the BFP reporter gene. After cleavage, the dissociative fragment was inverted to act as a homologous donor for gene repair, causing HEK293T cells to turn from blue to green (Appendix A). Editing outcomes in each cell were separated by fluorescence-activated cell sorting (FACS), and Illumina sequencing was used to determine the genes whose knockout leads to enrichment or depletion from each sorted population.

Under unperturbed conditions, this combination of reporter and CRISPR system yielded ~9.5% HDR (BFP edited to GFP) (Appendix A). To identify the genes involved in editing events, we used FACS to separate cells into unedited (BFP^+^) and HDR (GFP^+^) populations (Figure 2a). We recovered the crRNA abundance in each population by Illumina sequencing and compared these distributions to the edited unsorted populations to reveal which targeted gene increased (crRNA was enriched from the HDR population, but deleted from the BFP population) HDR activities. Our screening uncovered many genes promoting HDR (after crRNA editing), such as *SHROOM1* and *SLC36A3* (Figure 2b, Appendix A). *SHROOM1* belongs to *SHROOM* family and mainly involves in the assembly of microtubule arrays during cell elongation [38,39], SHROOM1 interacts with F-actin to ensure the development of neuroepithelial cells [40], and regulates gamma-tubulin distribution and microtubule architecture during epithelial cell shape change [41]. *SLC36A3* is belonging to *SLC36* family and responsible for amino acid and proton transport [42,43].

We designed specific siRNAs for the top eight genes whose crRNAs were enriched in the HDR population in order to validate the results of our screening. These siRNA showed efficient suppression to corresponding genes in HEK293T cells (Appendix A). Each siRNA was co-transfected with SpCas9 and sgRNA into the dual-cut BFP reporter cells, and the relative HDR ratio was calculated compared to NC by FACS. Four siRNAs exhibited obviously increased HDR efficiency, especially *SHROOM1* (Figure 2c and Appendix A). An additional three siRNAs for each effective gene were used in dual-cut BFP reporter cells to exclude the interference of off-target genes that similarly improved the HDR ratio (Appendix A).

In order to determine the HDR enhancement was actually induced by gene editing, we combined the paired crRNAs of four genes that increased in validated results with reporter cells to construct monoclonal cell lines, as well as *AAVS1* as a control. Since the 20th day after construction of the cell line, the HDR ratio of each cell line with SpCas9 system incision was recorded every two days (Figure 2d). The time curve showed that the HDR ratio of all cell lines experienced a 1.5-fold elevation compared to *AAVS1* from day 24 (Figure 2d). Genome sequencing analysis indicated that the targeting sites of crRNA pairs were edited efficiently (Appendix A). Apparent deletions around the two targets occurred in the *SHROOM1* paired crRNA expressed cell line (Appendix A). We mixed all 30-day monoclonal cell lines in a 1:1 ratio to simulate a mini-screening to find the one most enriched in HDR activity. As a result, the *SHROOM1* crRNA was enriched 4.7-fold more than *AAVS1* and therefore *SHROOM1* was selected as a potent suppressor of HDR in these reporter cells (Figure 2e).

### 2.3. Knockdown of SHROOM1 Can Promote In Vitro Genome Editing Efficiency

We examined whether knockdown of *SHROOM1* led to a more robust knock-in compared with undisturbed cells in vitro. To test this idea, we compared the HDR efficiency using siRNA as well as three types of donors: a single-strand donor (ss donor), a double-strand PCR donor (ds donor), and double-cut sites containing a plasmid donor (dc donor) (Figure 3a). All donors contained 800 bp left and right homologous arms. To evaluate the knock-in efficiencies, we aimed to fuse a P2A-EGFP reporter gene to the last codon of the *FBL* gene in HEK293T cells. The resulting HDR efficiencies were presented as a percentage of GFP^+^ cells (Figure 3a,b). At seven days after cell sorting HEK293T with an ss donor, sgRNA, and spCas9, the HDR efficiency of *SHROOM1* siRNA (3.65% ± 0.10%) was higher than NC siRNA (2.37% ± 0.05%) (Figure 3c). Genotyping showed that *SHROOM1* knockdown- mediated gene knock-in represented precise in-frame integrations, in common with the control (Appendix A). Similarly, using a ds donor, the HDR efficiency of *SHROOM1* decreased (6.66% ± 0.31%) but was still more than 3-fold higher than that of the NC group (2.19% ± 0.23%) (Figure 3c). HDR efficiency mediated by double-cut sites containing a plasmid donor was, remarkably, 10-fold higher than with classical donors [20,21]. In our study, *SHROOM1* knockdown elevated HDR efficiency (though it was already at a high level) when a dc donor was used, with the knock-in efficiency increasing from 19.3% ± 1.55% to 27.5% ± 1.95% (Figure 3c). The similar elevated HDR ratios were generated by another siRNA of *SHROOM1* (si*SHROOM1’*) (Appendix A). Therefore, *SHROOM1* knockdown can significantly promote HDR efficiency at the *FBL* locus, whichever donor is applied (Figure 3d).

We next examined HDR efficiency at the *LMNA* gene locus in HEK293T (Appendix A). When *SHROOM1* was knocked down by siRNA, the HDR efficiency, mediated by the ds donor, increased from 29.5% ± 2.00% to 36.8% ± 1.71%, and the HDR efficiency rose from 43.9% ± 2.26% to 51.3% ± 1.43% using a dc donor (Figure 3e). In human colorectal cancer cell line HCT116, the efficiency of *SHROOM1* knockdown was higher than in the control when a ds donor was used at the *FBL* or *LMNA* locus (Figure 3e). The same elevation of HDR efficiency was observed at the *Actb* locus in mouse hepatoma cell line Hepa1-6 with different types of donors when *SHROOM1* was inhibited (Figure 3f).

Together, these results indicated that the *SHROOM1* knockdown method yielded a higher HDR efficiency in multiple cell lines, regardless of which type of donor was used.

### 2.4. SHROOM1 Knockdown Can Promote Gene Integration in Mouse Embryos

To investigate whether the *SHROOM1* knockdown strategy could improve knock-in efficiency in generating gene-modified mice, we used the Cas9-Avidin/biotin-donor DNA system [19] and siRNA in mouse zygotes (Figure 4a and Appendix A). This system has been confirmed to result in a remarkable knock-in efficiency in mouse zygotes. The high affinity of streptavidin and biotin could promote accessibility of Cas9 and the DNA donor linked to them, then local concentration of the donor at the cutting site increased and which resulted in enhanced HDR efficiency. The injected zygotes were transferred to pseudo-pregnant mice for embryo development. Genotyping of new-born mice showed that knockdown of *SHROOM1* by 13.3 pg siRNA per zygote could improve precise integration efficiency from 0% to 6.5% at the *Ddx4* locus (Figure 4b,d and Appendix A). A similar improvement happened at the *Icos* locus, from 6.7% to 12.5%, with the same operation (Figure 4c,d and Appendix A). Interestingly, the imprecise integration (only at the 5′ junction or 3′ junction) efficiencies of these locus also increased, accompanied by precise integration (Figure 4b,c and Appendix A). Therefore, knockdown of *SHROOM1* could significantly promote gene integration in mouse embryos.

### 2.5. SHROOM1 Is a Potent Suppressor of HDR in Cells

We next explored whether the elevated knock-in efficiency was a result of the increased HDR induced by *SHROOM1* inhibition. Knockout of *SHROOM1* in the HEK293T cell line was constructed by SpCas9 and two sgRNAs targeting exon 4 of the primary gene transcript (Figure 5a). Genotyping showed that there was an 815 bp deletion and a frameshift (+1 bp) in this KO cell line (Figure 5a and Appendix A). Expression of SHROOM1 was totally abolished in KO cells (Figure 5b). Using the same knock-in strategy with a dc donor at the *FBL* locus, the knock-in efficiency of KO cells (38.2% ± 1.61%) was twice as high as in wild-type cells (19.4% ± 1.21%) (Figure 5c). When YU238259, an HR inhibitor [44], was added to treat KO cells during transfection, the knock-in efficiency dropped to 23.1% ± 2.21%. The efficiency was not significantly different in KO cells treated with Scr7 (a NHEJ inhibitor [22]) (Figure 5c). After cutting by CRISPR/Cas9 and sgRNA without a donor, NHEJ was generated in HEK293T cells and could be inhibited by Scr7 (Appendix A). But there is no difference of NHEJ frequency in *SHROOM1* KO cells comparing with wildtype cells. Re-expression of SHROOM1 in KO cells could visibly eliminate the enhancement of knock-in efficiency (Figure 5b,c). Therefore, deletion of *SHROOM1* caused a rise in knock-in efficiency through promoting HDR activity rather than NHEJ.

## 3. Discussion

In summary, through a genome-wide screening and following paired crRNA library screening with CRISPR-Cpf1, we have identified *SHROOM1* as a potent suppressor of HDR in BFP reporter cells. Knockdown of *SHROOM1* can improve the knock-in efficiency in both cultured mammalian cells and mouse embryos.

Genome-wide screening in iDR-GFP U2OS by siRNAs exhibits high stability and accuracy. Many genes appear in our sight, such as *CtIP* and *BRCA1*, and may play an important role in HDR maintaining activity. In order to exclude potential false positive and find potent HDR-suppressing genes, we use CRIPSR/Cpf1 and paired crRNAs to implement the next screening. Considering the instability and lower HDR incident of the BFP-reporter after transfecting the CRISPR system in U2OS, we designed a dual-cut BFP reporter in HEK293T to carry on the subsequent screening. Finally, we found suppressing *SHROOM1* could improve the HDR ratio in dual-cut BFP 293T significantly. In common with *SHROOM1*, inhibition of *SLC36A3* shows similar ability to enhance the HDR efficiency in siRNA-treated cells (Figure 2c) or stable cell line expressing paired crRNA and Cpf1 (Figure 2d). But the parallelly increased ratios may not completely represent the ability to promote HDR, because it was generated in cells with heterogeneity when transfected and background HDR frequency and edited for two days. Higher statistical stability of *SHROOM1* (5.55 ± 1.53, *p* = 0.016) shown in CRISPR-Cpf1 screening (Figure 2b and Appendix A) may cause remarkable enrichment in the scaled-down screening lasting for seven days comparing to *SLC36A3* KD (6.74 ± 3.70, *p* = 0.041) (Figure 2e).

Our work shows that HDR inhibition or re-expression of *SHROOM1* can eliminate the improvement of knock-in efficiency in KO cells, which indicates that deletion of *SHROOM1* allows cellular HDR activities to raise the gene editing efficiency. Though NHEJ is the main damage repair approach [45], it is not suitable for use with *SHROOM1*. In *SHROOM1*-KO HEK293T cells, the NHEJ frequency shows no difference with wildtype. *SHROOM1* has been reported to involve in the assembly of microtubule arrays during cell elongation. But the mechanism of *SHROOM1* regulating HDR still remains unknown. Side scatter (SSC) represents the complexity of the cell in the flow cytometer, and forward scatter (FSC) represents cell size. We found that deficiency of *SHROOM1* showed higher SSC and FSC value, which represents increased intracellular complexity and cell size of KO cells (Appendix A). This is a foreseeable result because of the function of *SHROOM1* in the cytoskeleton. We supposed the altered complexity and size might affect the cell cycle to regulate HDR activity which occurs at the late S/G2 phase. Wildtype and *SHROOM1* KO HEK293T were stained with propidium iodide (PI, a DNA dye which fluorescence is direct interrelated with binding DNA amount) and then analyzed by the flow cytometer to detect the possible change of cell numbers at different phases. It is depressing that there is no difference of cell cycle when *SHROOM1* is absent (Appendix A). Therefore, other methods like target searching or interaction analysis to find the functional pathway may provide some clues to uncover the mechanism that *SHROOM1* negatively regulates HDR.

Precision genome editing by modified nucleases such as the CRISPR-Cas9 system can be enhanced by the optimization of the donor, modulating the cellular DNA repair machine or selection of damage repair approaches. We demonstrated that inhibition of *SHROOM1* can promote cellular knock-in efficiency using different types of donors. Compared with other strategies such as expressing an HDR-promoted protein, adding the siRNA of *SHROOM1* has more advantages because of its smaller size and low cost. Altogether, our work establishes that inhibiting *SHROOM1* is a robust tool for enhancing CRISPR-mediated precision genome editing in human cells and mouse embryos, thereby aiding the study of gene function and modeling of human diseases.

## 4. Methods

### 4.1. Construction of Plasmids

To construct the rtTA-expressing lentiviral vector, an EF1α promoter, rtTA, and the IRES-connected neomycin were subcloned into the cloning site between cPPT/CTS and 3′ LTR of a modified pLKO vector (no. 13425, Addgene, Watertown, MA, USA).

To construct the I-*Sce*I induced expressing lentiviral vector, TRE-miniCMV-I-*Sce*I expression cassettes and a PGK-puromycin expression cassette were subcloned into the cloning site between cPPT/CTS and 3′ LTR of a modified pLKO vector (no. 13425, Addgene, Watertown, MA, USA).

To construct the HDR reporter lentiviral vector, CMV-SceGFP expression cassettes, a PGK-hygromycin expression cassette, and an interval GFP fragment were subcloned into the cloning site between cPPT/CTS and 3′ LTR of a modified pLKO vector (no. 13425, Addgene, Watertown, MA, USA).

To construct the Cpf1 expression lentiviral vector, a CMV-Cpf1-P2A-puromycin expression cassette was subcloned into the cloning site between cPPT/CTS and 3′ LTR of a modified pLKO vector (no. 13425, Addgene, Watertown, MA, USA).

To construct the dual-cut BFP reporter lentiviral vector, an EF1α-BFP expression cassette, WPRE, and truncated GFP were subcloned into the cloning site between cPPT/CTS and 3′ LTR of a modified pLKO vector (no. 13425, Addgene, Watertown, MA, USA).

To generate a single paired crRNA expression lentiviral vector, a U6 promoter, two B*smB*I sites, and a CMV-mcherry expression cassette were subcloned into the cloning site between cPPT/CTS and 3′ LTR of a modified pLKO vector (no. 13425, Addgene, Watertown, MA, USA).

To construct the double-cut sites donor for the *FBL*/*LMNA*/*Actb* gene, donor DNA (800 bp HAL-p2A-mGFP-800bp HAR) sandwiched by a 23-nt *Actb*-sgRNA target sequence, U6-*Actb*-sgRNA expression cassette, and CMV-mcherry expression cassette was subcloned into the cloning site of a modified pcDNA3.1+ vector (V79020, Invitrogen-Thermo Fisher Scientific, Waltham, MA, USA).

To construct the *SHROOM1* expression vector, *SHROOM1* cDNA was subcloned into the cloning site of a pcDNA3.1+ vector (V79020, Invitrogen-Thermo Fisher Scientific, Waltham, MA, USA).

Primary gene and donor sequences are given in Appendix A.

### 4.2. Cell Culture and Cell Lines

U2OS, HEK293T, HCT116, and Hepa1-6 cells were obtained from the American Type Culture Collection (ATCC, Manassas, VA, USA). Cell lines were cultured in DMEM supplemented with 10% fetal bovine growth serum (SH30084.03, Hyclone-Thermo Fisher Scientific, Waltham, MA, USA) and 1× penicillin-streptomycin (10378016, GIBCO-Thermo Fisher Scientific, Waltham, MA, USA). Cells were grown at 37 °C with 5% CO_2_. To generate cell lines expressing the induced DR-GFP reporter, U2OS cells were transduced with lentiviruses (multiplicity of infection [MOI] = 0.1) carrying the EF1α-rtTA expression cassette, TRE-miniCMV-I *Sce*I expression cassette, and CMV-SceGFP expression cassette successively. G418 (1 mg/mL), puromycin (1.25 μg/mL), or hygromycin (1 mg/mL) was used to select the corresponding stable cells. For generating dual-cut BFP reporter cells, HEK293T cells were transduced with lentiviruses (multiplicity of infection [MOI] = 0.1) carrying the CMV-Cpf1-P2A-puromycin expression cassette, then selected by puromycin (1.25 μg/mL). Subsequently, the stable cells were transduced with lentiviruses (multiplicity of infection [MOI] = 0.1) carrying a dual-cut BFP reporter and sorted by flow cytometry to produce a pure population of dual-cut BFP reporter cells (BFP-293T). To generate single paired crRNA expressed library cells, BFP-293T cells were transduced with lentiviruses (multiplicity of infection [MOI] = 0.1) to carry U6-paired crRNA and CMV-mcherry expression cassettes and sorted by flow cytometry.

### 4.3. Genome-Wide Screening

First, 2000 iDR-GFP U2OS cells were planted in 384-well plates. Eighteen hours later, 0.15 μL RNAiMAX (catalog 13778150, Invitrogen-Thermo Fisher Scientific, Waltham, MA, USA) and 2 pmol siRNAs (candidate or positive control, Genepharma, Suzhou, China) were transferred into each well by the Agilent (Santa Clara, CA, USA) Bravo automated liquid handling platform. Forty-eight hours after transfection, cells were treated with 2 μg/mL doxycycline (D8960, Solarbio, Beijing, China) for two days to induce DSB and HDR. Individual wells in plates were immobilized and stained by DAPI. Then, fluorescent data were collected by an Image Xpress instrument (Molecular Devices, San Jose, CA, USA) and processed by Meta Xpress. Ratio = 10 × No. (GFP^+^ cells)/No. (Total cells), FC = Ratio (siRNA)/Ratio (NC).

### 4.4. Genomic DNA Extraction and PCR Amplification for Editing Region

Seventy-five percent of the edited cells (~500,000 cells) were collected every two days from day 20 to day 38 after cell sorting and washed once in a PBS buffer solution. Genome extractions were carried out according to the instructions of the kit (DP304, Tiangen, Beijing, China) and eluted with 80 μL distilled water for downstream analysis. Amplification of edited loci was performed with the locus-specific primer pairs described in Appendix A using 2 × Q5 master mix (M0494L, New England Biolabs, Ipswich, MA, USA) and 200 ng of genomic DNA. The thermocycler was set for one cycle at 98 °C for 30 s, 35 cycles at 98 °C for 15 s, 60 °C for 15 s, and 72 °C for 10 s, respectively, and one cycle at 72 °C for 1 min, and held at 4 °C. PCR amplicons were run on a 1.5% agarose gel to verify the size and purity, and quantified by nanodrop. The resulting DNA was used for direct analysis or reamplified with primers containing Illumina adaptors.

### 4.5. Next-Generation Sequencing (NGS) Library Generation and Sequencing

One hundred nanograms of purified PCR amplicons were used as a library template. The sequencing library was obtained from the replicates using a NEBNext Ultra II RNA Kit (E7775S, New England Biolabs, Ipswich, MA, USA). Pooled samples were purified with SPRI beads. Library size and purity was verified by Agilent 2100 before sequencing on a Nova seq (Illumina, San Diego, CA, USA) using a Reagent Kit S2 (Novaseq 6000, Illumina) (2 × 150 bp).

### 4.6. Pooled Screen

After culturing for 28 days, one 1 × 10^7^ aliquot of single paired crRNA expressed library cells was collected for sequencing of library quality control, a second 1 × 10^7^ cell aliquot was collected for transfection, and the remaining cell libraries were frozen. Plasmid DNA (2 μg SpCas9 and 1.2 μg sgRNA) and lipofectamine 2000 (6 μL) were incubated together and transfected into 3 × 10^6^ cells cultured in a single well of a six-well plate. The transfected cell libraries were cultured for seven days. A 1 × 10^7^ cell aliquot was collected from the transfected cell library (Unsorted), and 2 × 10^7^ cells were sorted into BFP^+^ (Unedited) and GFP^+^ (HDR) populations. The collected cell populations were rinsed in PBS and frozen at −80 °C. DNA from each cell population—unsorted, unedited, and HDR—was purified with genome purification kits (DP304, Tiangen, Beijing, China) and the total amount of DNA was quantified. A maximum of 1 μg of genomic DNA was amplified in a single KAPA HiFi PCR (KK2602, KAPA Biosystems-Roche, Pleasanton, CA, USA) reaction using primers specific to the crRNA cassette. Up to 24 PCR reactions were set up for each cell population to obtain the desired coverage of the cell library. The thermocycler was set for one cycle of 98 °C for 30 s, 23 cycles of 98 °C for 20 s, 56 °C for 15 s, and 72 °C for 25 s, respectively, and one cycle of 72 °C for 5 min. PCR reactions were pooled and run on 1.5% agarose gel and purified by a gel extraction kit (28706, Qiagen, Germantown, MD, Germany). Amplified DNA from each cell population was normalized to input cell numbers and prepared for sequencing as mentioned above.

### 4.7. Pooled Screen Analysis

Sequence reads were trimmed, aligned to crRNA sequence templates, and quantified. Read counts for each crRNA pairs were normalized and compared to the distribution of untargeted control guides to determine the significance and log_2_ magnitude of change.

### 4.8. qPCR

For qPCR, 100,000–200,000 cells were collected and RNA extracted with the TRNzol (DP424, Tiangen Beijing, China). cDNA was produced from 1 μg of purified RNA using the FastKing RT Kit (KR116, Tiangen Beijing, China). qPCR reactions were performed with the SuperReal SYBR Green PreMix Plus (FP205, Tiangen Beijing, China) in a total volume of 10 μL, with primers at final concentrations of 500 nM. The thermocycler was set for one cycle of 95 °C for 15 min, and 40 cycles of 95 °C for 10 s, 60 °C for 20 s, and 72 °C for 32 s, respectively. Fold enrichment of the assayed genes over the control *GAPDH* loci was calculated using the 2^−ΔΔCt^ method.

### 4.9. siRNA Interference and HDR Assays with Dual-Cut BFP Reporter

Dual-cut BFP 293T cells were seeded at 60–70% confluency into 24-well plates. For qPCR, siRNA (500 ng, Appendix A) was transfected individually using lipofectamine 2000 (1 μL) and harvested two days after transfection. For HDR assays, a cocktail of siRNA (500 ng), SpCas9 (800 ng), and sgRNA (400 ng) was transfected using lipofectamine 2000 (2.5 μL). Enriched paired crRNA stably expressed dual-cut BFP 293T cells were seeded in the same manner described above and transfected with SpCas9 (800 ng) and sgRNA (400 ng) for HDR assays. The cells were collected three days after transfection for analyzing by flow cytometry for GFP^+^ cells using a LSRFortessa (Becton, Dickinson and Company, San Jose, CA, USA).

### 4.10. In Vitro Gene Knock-in

HEK293T, HCT116, Hepa1-6, and *SHROOM1* knockout HEK293T cells were seeded at 60–70% confluency into 12-well plates. For the ss donor or ds donor, these cells were transfected with donor (800 ng), SpCas9 (1200 ng), sgRNA (600 ng, Appendix A), and siRNA (1000 ng) using lipofectamine 2000 (8 μL). For the dc donor, cells were transfected with donor-sgRNA cassette (1200 ng), SpCas9 (1200 ng), and siRNA (1000 ng). Modified cells were collected two days after transfection and sorted by flow cytometry for mcherry^+^ cells using a MoFlo (Beckman Coulter, Brea, CA, USA). Seven days after sorting, the ratio of GFP^+^ cells was analyzed using a LSRFortessa (Becton, Dickinson and Company, San Jose, CA, USA) (Figure 4b). Genotyping of GFP+ cells was conducted using primers in Appendix A.

For the comparison of different knock-in strategies, treatment with 5 μM YU238259 (S8379, Selleck Chemicals, Houston, TX, USA) and 1 μM Scr7 (S7742, Selleck Chemicals, Houston, TX, USA) was started one day before transfection and was continued until two days after transfection. The *SHROOM1*-expressed plasmid (1000 ng) was co-transfected with the components above into KO cells.

### 4.11. Western Blotting

Primary antibodies against the following proteins were used: SHROOM1 (bs-13735R; Bioss, Beijing, China); GAPDH (sc-365062; Santa Cruz, Dallas, TX, USA). For each protein antibody, the manufacturer’s recommended dilutions were used. Mouse or rabbit immunoglobulin G was visualized with the following HRP-conjugated secondaries at a 1:5000 dilution: horse anti-mouse (# 7076S, Cell Signaling Technology, Danvers, MA, USA); goat anti-rabbit (# 7074S, Cell Signaling Technology, Danvers, MA, USA). The gray-scale value was analyzed by ImageJ software.

### 4.12. Micro-Injection and Genotyping

*Ddx4* and *Icos* targets were designed according to the protocol described at http://crispr.mit.edu. The cleavage efficiency was measured using in vitro detection assay (VK-007, ViewSolid, Beijing, China). The genomic sequences were amplified and purified, and then used as cutting templates for each sgRNA. For each locus, four targets were designed and their efficiencies measured, and the most efficient target was selected (Appendix A). All animal procedures were performed strictly according to the Animal Care Guidelines. Fertilized zygotes were collected from ICR mouse oviducts. For injection, Cas9-Avidin mRNA (100 ng/μL), sgRNA (20 ng/μL), biotin-ssDNA (20 ng/μL), and siRNA (1.33 μg/μL) (Appendix A) were mixed, and then injected into zygotes. The injected zygotes were first cultured in KSOM (M1450, Easycheck, Nanjing, China) with FBS at 37 °C and 5% CO_2_, and then transferred into pseudo-pregnant female ICR mice. Mouse tails were lysed at 55 °C with proteinase K overnight. Genomic DNA were extracted from F_0_ tails and amplified by MightyAmp (074A, Takara, Kusatsu, Shig, Japan). Two pairs of primers were designed for the 5′ junction and 3′ junction for detecting the precise gene insertion (Appendix A). After adding A at the 3′ terminal, they were ligased into a T vector for further sequencing.

### 4.13. Cell Cycle Analysis

Cells were seeded in a 6-well tissue culture plate (4 × 105 cells/well) and digested with 0.05% trypsin. After treatment, the cells were collected and washed with PBS. RNase A solution (100 μL) was added, and cells were incubated for 30 min at 37 °C. Finally, 400 μL PI (P8080-10 mg, Solarbio, Beijing, China) was added and incubated for 30 min at room temperature. The DNA content was detected by flow cytometry. The data were analyzed by LSRFortessa (Becton, Dickinson and Company, San Jose, CA, USA). The percentage of cells in the G1 phase, the S phase, and the G2 phase were analyzed.

## Figures and Tables

**Figure 1 ijms-21-05821-f001:**
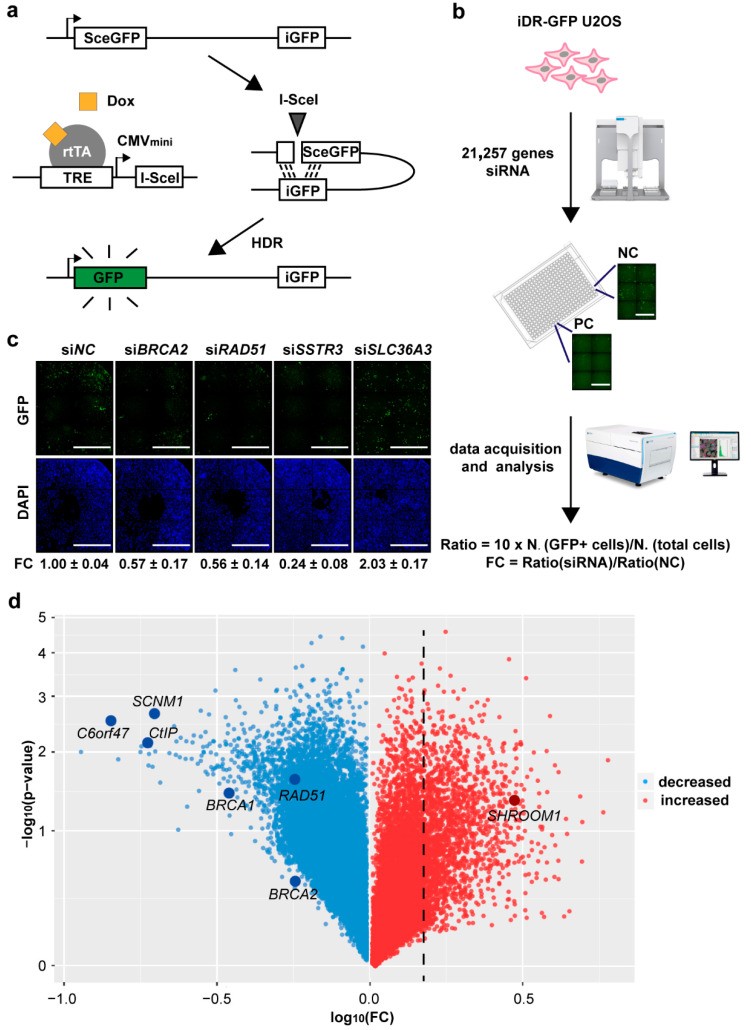
Genome-wide screening of HDR (homology directed repair)-modulated genes. (**a**) Schematic overview of the HDR progress in induced DR-GFP U2OS cells. Dox, doxycycline. (**b**) A flow diagram of the genome-wide screening using a high-content instrument. PC, positive control, *BRCA2* and *RAD51*; NC, negative control. Scale bar, 1000 μm. (**c**) Fluorescence pictures captured by a high-content instrument. FC, fold change. Every picture was stitched together from individual fields of vision. Scale bar, 1000 μm. (**d**) Multiple HDR-increased and decreased genes in the genome-wide screening. Representative genes are highlighted in dark blue. FC (fold change) of genes on dashed line equals 1.5. Data were generated from *n* = 3 independent experiments. The *p*-value was calculated using a two-sided Student’s *t*-test.

**Figure 2 ijms-21-05821-f002:**
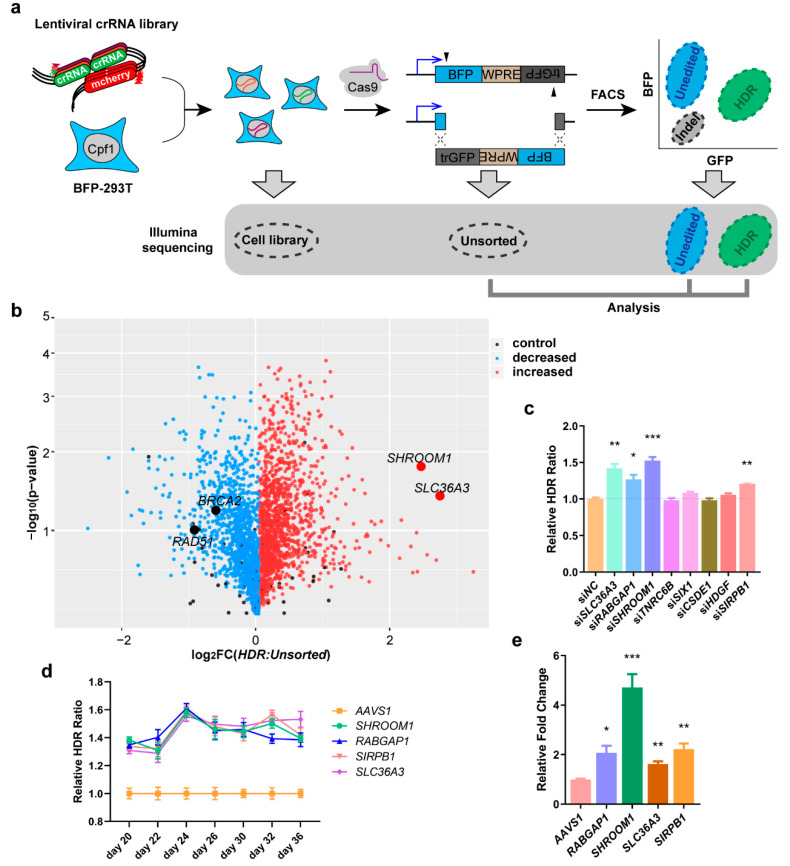
Paired crRNA library screening identified that *SHROOM1* is an HDR suppressor of dual-cut BFP (blue fluorescence protein) reporter. (**a**) Schematic of paired crRNA library screening strategy in the main text; (**b**) multiple genes were enriched in the HDR population. Representative genes are highlighted in red. Data were generated from *n* = 4 independent experiments. The *p*-value was calculated using a two-sided Student’s *t*-test. (**c**) Relative HDR ratio of dual-cut BFP reporter cells treated with individual siRNA of top eight genes enriched in the screening. Data were generated from *n* = 3 independent experiments. Error bars, ± SD * *p* < 0.05; ** *p* < 0.01; *** *p* < 0.001 by two-sided Student’s *t*-test. (**d**) Relative HDR ratio-time curve of dual-cut BFP reporter cell lines expressing mono-paired crRNAs. Data were collected every two days from day 20 to day 36 after cell sorting. Data were generated from *n* = 3 independent experiments. (**e**) Genes enriched in the HDR population of simulated screening. Data were generated from *n* = 4 independent experiments. Error bars, ±SD * *p* < 0.05; ** *p* < 0.01; *** *p* < 0.001 by two-sided Student’s *t*-test.

**Figure 3 ijms-21-05821-f003:**
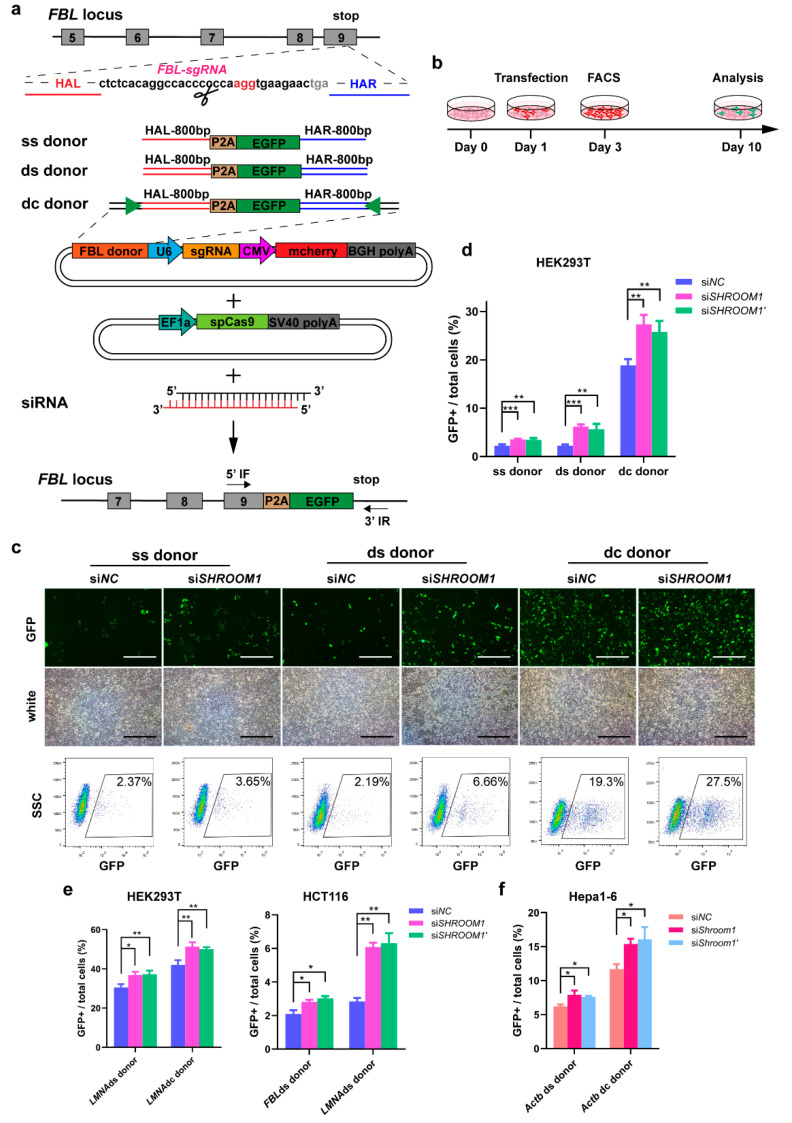
Knockdown of *SHROOM1* enhances the knock-in efficiency of cells in vitro. (**a**) Schematic overview of gene-targeting strategies with siRNA and different types of donor at the *FBL* locus. HAL/HAR, left/right homology arm; triangles, sgRNA target sites; IF/IR, inserted forward/reverse primer. ss, single strand; ds, double strand; dc, double cut. (**b**) Experimental scheme for targeted *FBL*-2A-GFP knock-in in HEK293T cells. Representative visual fields and sorting charts (**c**) and relative knock-in efficiency. Scale bar, 200 μm. (**d**) of ss, ds, and dc donor-based strategies with *SHROOM1* siRNA or not at the *FBL* locus in HEK293T cells. Data were generated from *n* = 3 independent experiments. Error bars, ± SD ** *p* < 0.01; *** *p* < 0.001 by two-sided Student’s *t*-test (**e**,**f**). Relative knock-in efficiency of ds and dc donor-based strategies with siRNA in HEK293T, HCT116, or Hepa1-6 cells. Data were generated from *n* = 3 independent experiments. Error bars, ± SD * *p* < 0.05; ** *p* < 0.01 by two-sided Student’s *t*-test.

**Figure 4 ijms-21-05821-f004:**
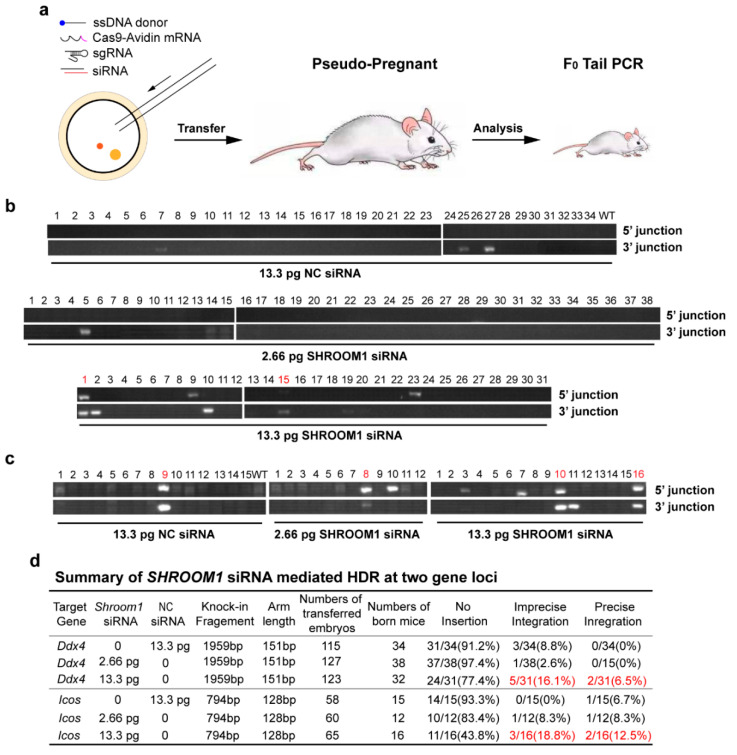
Knockdown of *SHROOM1* enhances the knock-in efficiency of mouse embryos. (**a**) Experimental design of micro-injection. Cas9-Avidin mRNA, sgRNA, biotin-ss donor, and siRNA were injected into mouse zygotes and the injected zygotes were transferred to pseudo-pregnant mice for genotyping analysis. Genotyping of *Ddx4* locus (**b**) and *Icos* locus (**c**) in mice treated with *SHROOM1* siRNA or NC siRNA after incision by CRISPR/Cas9; (**d**) Summary of *SHROOM1* siRNA-mediated HDR at the *Ddx4* and *Icos* loci. Results with significant differences are highlighted in red.

**Figure 5 ijms-21-05821-f005:**
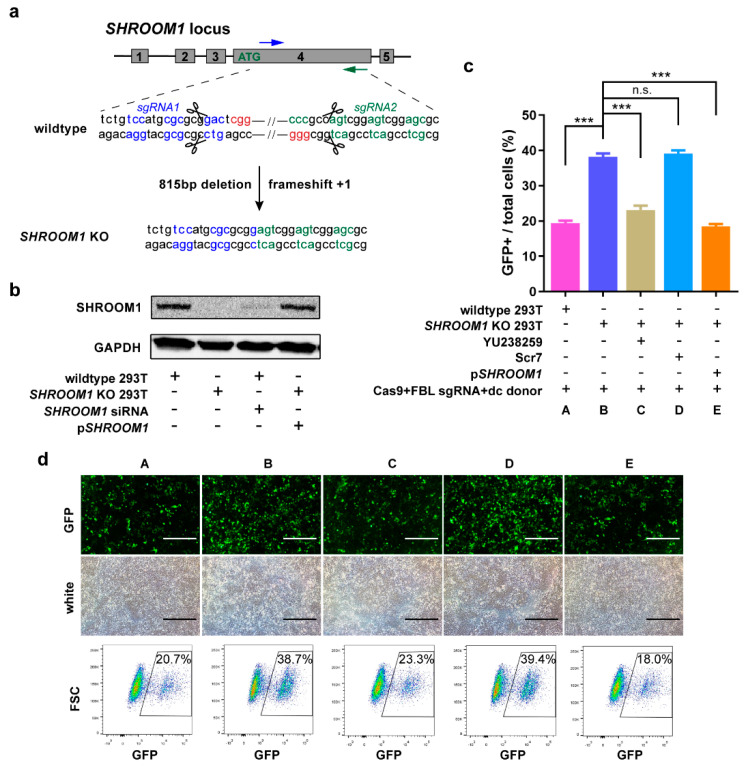
*SHROOM1* is a potent suppressor of HDR progress. (**a**) Schematic of *SHROOM1* deletion using CRISPR/Cas9 and two sgRNAs in HEK293T cells. PAM, highlighted in red; protein codon, highlighted in blue or green, KO for knockout; (**b**) Western blot of different types of cells or treatment. KO, for knockout; p*SHROOM1* for *SHROOM1* cDNA contained plasmid; relative knock-in efficiency (**c**) and representative visual fields and sorting charts (**d**) in *SHROOM1* knockout or wild-type HEK293T cells with treatments. YU238259, an HR inhibitor; Scr7, a NHEJ inhibitor; dc, double-cut sites contained donor. Data were generated from *n* = 3 independent experiments. Error bars, ± SD *** *p* < 0.001; n.s., no significance; by two-sided Student’s *t*-test. Scale bar, 200 μm.

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
