# Peer review of "Suppression of SHROOM1 Improves In Vitro and In Vivo Gene Integration by Promoting Homology-Directed Repair"

_ijms, 2020, doi:10.3390/ijms21165821_

Round 1
Reviewer 1 Report
Review on Zhao et al.
The paper by Zhao et al. describes the identification and characterization of a gene which regulates homologous recombination (HR), thus, HR-mediated DNA repair such as gene knock-in (targeting). By using CRISPR/Cas9-induced HR fluorescent report system(s), the authors screened the siRNA which decrease or increase HR frequencies, and identified one gene, SHROOM1. Moreover, the knockdown of SHROOM1 in human and mouse cells as well as mouse zygotes showed an increased targeting frequency relative to control knockdown. Further study shows that knockout cell lines for SHROOM1 also elevated knock-in frequency. The works in the paper had done in a reasonable quality. Moreover, the discovery of a new HR negative regulator is of interest in researchers in the field of HR and also in genome editing. The results are worthwhile in publication. However, several concerns described below should be addressed. Particularly, more detailed explanation on the methods in the main text and data deposition for the screening results should be essential for the publication.
In the manuscript, it is very surprising that, in the main text, the authors did not describe what SHROOM1 (and also the other candidate, SLC36A3) protein is and what kind of functions have been described in the past. And also, poor discussion on how SHROOM1 negatively regulates HR is in Discussion.
Major points:
- It is very important for the authors to describe their GFP-based recombination system (Figure 1) and to validate it in a quantitative way. Because of poor description on the system, it is hard to evaluate. For example, in Figure 1c, the authors presented only relative ratio called FC. However, absolute values (frequencies of GFP-positive cells should be shown with before/after addition of tetracycline) should be presented in the text or Figures (or Supplementary). Moreover, FC should be associated with statistical error, which should be shown in Supplement (see #2).
- All FC values (and real frequencies, SD, P-values) of 21,257 gene knockdown (KD) in the screening described in Figure 1 should be presented in Supplementary Tables.
- In screening described in Figure 2, control experiments using RAD51 and BRCA2 should be included.
- As pointed out in #2, all values should be shown in Supplementary tables.
- In Figure 2b, c and d, SHROOM1 and SLC36A3 show similar increased HR efficiency. However, in Figure 2e, SLC36A3 KD shows weaker increased relative to SHROOM1 KD. It seems that the authors are used similar measurement of four independent experiments. How the difference is generated? Please explain it in the text
- What is a frequency of NHEJ in SHROOM1-KD cells? The authors just used the assay in Figure 3 etc to measure it (simple junction analysis after the transfection of CRISPR/Cas.
Minor points:
- Page 2, line 5-6 from the bottom: “in addition, the fluorescent brightness of randomly selected genes, SSTR3 and SLC36A3, matched the reading data well.” In Figure 1c, the values of SSTR3 and SLC36A knockdown are 0.29 and 2.1, respectively. It is hard to catch what the authors mean in this sentence. Please rephrase.
- Figure 1d; where are RAD51 and SHROOM1 in this plot?
- Page 4, line 2; “37 controls” were used in experiments described in Figure 2. What are these controls? Please indicate what genes were selected in Supplement.
- Supplementary Figure 2a; Need more words in legend. It is impossible to follow what an object in third raw in the Figure represents.
- Supplementary Figure 2b; Please indicate the size of PCR products in before/after editing.
- Supplementary Figure 2c; R1/2, repeat1/2 means independent experiments? Please clarify it. And, moreover, in 28D-R1, there are similar edited products compared to that in 0D-R1. This means very inefficient cutting by CRISPR/Cas for this system? Please explain in more precisely.
- Supplementary Figure 2d; Is this a sequence around crRNA1 or -2? How the sequence of the products was determined? For shorter PCR products? Since the experiment used paired crRNA (Fig. S2b) with a shorter product (S2c), why the authors detected alternation of a single site rather than deletion?
- Supplementary Figure 3a and b, page 4, third paragraph; What reporter assay was used to measure HR frequencies?
- Page 4, line 7 from the bottom; “0.5” fold means half reduction. Please replace it with “1.5” fold or by 50% increase etc.
- Page 8, line 2; Please explain Cas9-avidin-biotin-donor DNA system in more detail for readers.
Author Response
Dear Reviewer:
Thank you for your comments concerning our manuscript entitled “Suppression of SHROOM1 Improves In Vitro and In Vivo Gene Integration by Promoting Homology Directed Repair” (ID: ijms-862174). Those comments are all valuable and very helpful for revising and improving our paper, as well as the important guiding significance to our researches. We have studied comments carefully and have made correction which we hope meet with approval. Revised portion are marked in red in the paper. The main corrections in the paper and the responds to the comments are as flowing:
Responds to the reviewer’s comments:
Comments: In the main text, the authors did not describe what SHROOM1 (and also the other candidate, SLC36A3) protein is and what kind of functions have been described in the past. And also, poor discussion on how SHROOM1 negatively regulates HR is in Discussion.
We are sorry for our negligence of not describing SHROOM1 and SLC36A3 in the main text. We have added description of SHROOM1 and SLC36A3 as below to the end of paragraph 2 on page 4. “SHROOM1 belongs to SHROOM family and mainly involves in the assembly of microtubule arrays during cell elongation [38, 39], SHROOM1 interacts with F-actin to ensure the development of neuroepithelial cells [40], and regulates gamma-tubulin distribution and microtubule architecture during epithelial cell shape change [41]. SLC36A3 is belonging to SLC36 family and responsible for amino acid and proton transport [42, 43].”
We have added following discussion of how SHROOM1 negatively regulate HR to paragraph 3 on page 10. “SHROOM1 has been reported to involve in the assembly of microtubule arrays during cell elongation. But the mechanism of SHROOM1 regulating HDR still remains unknown. Side scatter (SSC) represents the complexity of cell in flow cytometer, and Forward scatter (FSC) represents cell size. We found that deficiency of SHROOM1 showed higher SSC and FSC value which represents increased intracellular complexity and cell size of KO cells (Figure S6c). This is a foreseeable result because of the function of SHROOM1 in cytoskeleton. We supposed the altered complexity and size might affect cell cycle to regulate HDR activity which occurs at late S/G2 phase. Wildtype and SHROOM1 KO HEK293T were stained with Propidium Iodide (PI, a DNA dye which fluorescence is direct interrelated with binding DNA amount) and then analyzed by flow cytometer to detect the possible change of cell numbers at different phases. It is depressing that there is no difference of cell cycle when SHROOM1 is absent (Figure S6d). Therefore, other methods like target searching or interaction analysis to find functional pathway may provide some clues to uncover the mechanism that SHROOM1 negatively regulates HDR.”
Major point 1: It is very important for the authors to describe their GFP-based recombination system (Figure 1) and to validate it in a quantitative way. Because of poor description on the system, it is hard to evaluate. For example, in Figure 1c, the authors presented only relative ratio called FC. However, absolute values (frequencies of GFP-positive cells should be shown with before/after addition of tetracycline) should be presented in the text or Figures (or Supplementary). Moreover, FC should be associated with statistical error, which should be shown in Supplement (see #2).
Thanks a lot for this suggestion. We added the description on page 2, “After treating with siRNA or NC and inducing by Dox, in each field, cells with the HDR activity which exhibited green fluorescence would be counted as N. (GFP+ cells). While total cells were stained with 4',6-diamidino-2-phenylindole (DAPI) and counted as N. (total cells) (Figure S1c). Six adjacent fields from a same sample were collected, and final HDR-ratio would be calculated by the ratio of N. (GFP+ cells) to N. (total cells). We used relative FC of HDR-ratio to define the ability of a gene affecting HDR (Figure 1b).”
On page 2, all FC values, real frequencies, SD and P-values are presented in the supplementary table 4. The corresponding statistical errors are added to associate with FC in Figure 1c, and all frequencies and their statistical errors are shown in supplementary table 4.
Major point 2: All FC values (and real frequencies, SD, P-values) of 21,257 gene knockdown (KD) in the screening described in Figure 1 should be presented in Supplementary Tables.
Thank for your suggestion. All FC values, real frequencies, SD and P-values are presented in the supplementary table S4 and noted on page 2, paragraph 4, line 15.
Major point 3: In screening described in Figure 2, control experiments using RAD51 and BRCA2 should be included.
Thanks. RAD51 and BRCA2 were included in our CRISPR screening (see supplementary table 5). We have marked the location of them in Figure 2b on page 5. Because subsequent validation and knock-in assays are focused on HDR-supressed genes, NC siRNA or AAVS1 may be more appropriate to be set as a control.
Major point 4: As pointed out in #2, all values should be shown in Supplementary tables.
Thank for your notice. All values are shown in supplementary table 5.
Major point 5: In Figure 2b, c and d, SHROOM1 and SLC36A3 show similar increased HR efficiency. However, in Figure 2e, SLC36A3 KD shows weaker increased relative to SHROOM1 KD. It seems that the authors are used similar measurement of four independent experiments. How the difference is generated? Please explain it in the text
We measure HDR frequency in Figure 2c and d by using flow cytometer two days after incising by CRISPR-Cas9, Figure 2c was in dual-cut BFP cells treated with siRNA while Figure 2d in dual-cut BFP cells stable expressing paired crRNAs and Cpf1. We measured enrichment of crRNA in GFP positive cells which occurred HDR in Figure 2b and e.
We have added following explanation for the generated difference in discussion section on page 10. “In common with SHROOM1, inhibition of SLC36A3 shows similar ability to enhance the HDR efficiency in siRNA-treated cells (Figure 2c) or stable cell line expressing paired crRNA and Cpf1 (Figure 2d). But the parallelly increased ratios may not completely represent the ability to promote HDR, because it was generated in cells with heterogeneity when transfected and background HDR frequency and edited for two days. Higher statistical stability of SHROOM1 KD (5.55±1.53, p = 0.016) shown in CRISPR-Cpf1 screening (Figure 2b and Table S5) may cause remarkable enrichment in the scaled-down screening lasting for seven days comparing to SLC36A3 KD (6.74 ± 3.70, p = 0.041) (Figure 2e)”.
Major point 6: What is a frequency of NHEJ in SHROOM1-KD cells? The authors just used the assay in Figure 3 etc to measure it (simple junction analysis after the transfection of CRISPR/Cas.
We have measured the frequency of NHEJ in SHROOM1-KD cells (Figure S6b). SHROOM1-KD cells and wildtype HEK293T cells treated with Scr7 or not were transfected with Cas9 and sgRNA targeting to EMX1 or FBL locus. Then the corresponding edited genomes were sequenced and the indels ratio was used to represent NHEJ frequency. Our results show that there is no difference of NHEJ frequency in SHROOM1-KD cells.
We have indicated the following description and note of supplementary Figure 6b in paragraph 1 on page 9. “After cutting by CRISPR/Cas9 and sgRNA without donor, NHEJ was generated in HEK293T cells and can be inhibited by Scr7 (Figure S6b). But there is no difference of NHEJ frequency in SHROOM1 KO cells comparing with wildtype cells.”
Minor point 1: Page 2, line 5-6 from the bottom: “in addition, the fluorescent brightness of randomly selected genes, SSTR3 and SLC36A3, matched the reading data well.” In Figure 1c, the values of SSTR3 and SLC36A knockdown are 0.29 and 2.1, respectively. It is hard to catch what the authors mean in this sentence. Please rephrase.
Thank you for your suggestion. We have rephrased this sentence with the following description in line 5-6 from the bottom on page 2. “In addition, the fluorescence brightness of two randomly selected genes (SSTR3 and SLC36A3) matched the reading data well, which represented high accuracy of our screening”.
Minor point 2: Figure 1d; where are RAD51 and SHROOM1 in this plot?
Thanks. These genes are now marked in Figure 1d on page 3.
Minor point 3: Page 4, line 2; “37 controls” were used in experiments described in Figure 2. What are these controls? Please indicate what genes were selected in Supplement.
Thanks. We added 37 controls in sheet 2 of supplementary table 5.
Minor point 4: Supplementary Figure 2a; Need more words in legend. It is impossible to follow what an object in third raw in the Figure represents.
Thanks. We made modification of supplementary Figure 2a and added more description below in the legend of Figure S2 on page 2. “Two crRNAs targeting promoter or splicing sites (1) were paired randomly and combined with DR sequence to form an oligo (2). The pooled oligo array (3) consisted of 11918 oligos from 1347 candidates and 37 controls was used to amplified PCR and then integrated into a pLKO vector to form the plasmid library (4)”.
Minor point 5: Supplementary Figure 2b; Please indicate the size of PCR products in before/after editing.
Thanks. The sizes of PCR products are shown in supplementary Figure 2c and indicated in legend of Figure S2c on page 2.
Minor point 6: Supplementary Figure 2c; R1/2, repeat1/2 means independent experiments? Please clarify it. And, moreover, in 28D-R1, there are similar edited products compared to that in 0D-R1. This means very inefficient cutting by CRISPR/Cas for this system? Please explain in more precisely.
Thanks. Repeat1/2 means two independent experiments in HEK293T cells expressing one copy of paired UPF1 crRNAs and integrated Cpf1. The following description has been added to legend of Figure S2c on page 2. “genotyping by detected primers in (b) and editing efficiencies of two independent HEK293T clones expressing one copy of paired UPF1 crRNAs and integrated Cpf1 at day 0 and day 28; A edited band (about 621 bp) was generated from wildtype band (889 bp) after incision induced by Cpf1 and crRNAs. D0, the day of cell sorting after infection of the paired crRNA lentivirus; D28, twenty-eight days after cell sorting and lentiviral infection; R1 or R2, independent experimental repeat 1 or 2”.
We are sorry for not showing editing efficiency in Figure S2c. As indicated in Figure S2c, after editing for twenty-eight days by Cpf1 and paired crRNAs, deletion ratio of repeat 1 comes to 18.2% comparing to 1.5% at day 0. Another independent experimental repeat (R2) shows similar deleted efficiency after cutting (51.2% versus 4.0%). The efficiencies of deletion are calculated by grey value using ImageJ software. Considered the sequencing results of region between two crRNA targets in Figure S2c, we think our CRISPR/Cpf1 system possesses high editing efficiency. Corresponding deletion ratios are indicated in supplementary Figure 2c.
Minor point 7: Supplementary Figure 2d; Is this a sequence around crRNA1 or -2? How the sequence of the products was determined? For shorter PCR products? Since the experiment used paired crRNA (Fig. S2b) with a shorter product (S2c), why the authors detected alternation of a single site rather than deletion?
The sequence in Supplementary Figure 2d contains crRNA 1, crRNA 2 and regions between them. “240 bp” stands for omit of intermediate region. We have added following annotation to the legend of supplementary Figure 2d on page 2. “sequencing results of the cocktail of shorter (621 bp) and longer (889 bp) bands of D28-R1 and D28-R2 in Figure S2c; Triangles, deletion of nucleotides; crRNA 1 sequence, highlighted in green, crRNA 2, highlighted in blue; PAM, highlighted in grey; hollow triangle, deletion of nucleotides;”
There may exist three types of editing results induced by paired crRNA (Figure S2b) and CRISPR Cpf1: no editing (A in Figure S2d), one cutting by crRNA 1 or 2 (B, C and D in Figure S2d) and deletion (E and F in Figure S2d). We detected the deletion band and no size-changed band and mixed them to sequence. The sequencing results show high editing efficiency (23/30) because a single cutting perhaps affects the expression of a gene. We have revised the supplementary Figure 2d and added indication to legend of Figure S2d on page 3. Meanwhile, we add following description of this detection in page 4, paragraph 1, line 8. “Genotyping and sequencing results showed that, after 28 days, the single paired crRNA copy of UPF1 could guide Cpf1 to generate effective editing including single cutting and long fragment deletion”.
Minor point 8: Supplementary Figure 3a and b, page 4, third paragraph; What reporter assay was used to measure HR frequencies?
We used HEK293T cells to detect the knockdown efficiency of siRNAs, and the qPCR results of mRNA are shown in supplementary Figure 3a. We have added following description to page 4, paragraph 3, line 2. “These siRNA showed efficient suppression to corresponding genes in HEK293T cells”.
The dual-cut BFP reporter cells (BFP-293T) were used to measure HR frequencies of supplementary Figure 3b (now Figure S3c in revised edition). The HR frequencies were detected by flow cytometer after treated with siRNA and Cpf1/crRNA. We have added “in dual-cut BFP reporter cells” to indicate which reporter we used in page 4, paragraph 3, line 6.
Minor point 9: Page 4, line 7 from the bottom; “0.5” fold means half reduction. Please replace it with “1.5” fold or by 50% increase etc.
Thank you. The corrected expression is revised in page 4, line 7 from the bottom.
Minor point 10: Page 8, line 2; Please explain Cas9-avidin-biotin-donor DNA system in more detail for readers.
We have added detailed description as below in page 8, paragraph 1, line 3.
“This system has been confirmed to result a remarkable knock-in efficiency in mouse zygotes. The high affinity of streptavidin and biotin could promote accessibility of Cas9 and DNA donor linked to them, then local concentration of donor at cutting site increased and which resulted enhanced HDR efficiency.”
Special thanks to you for your good comments.

Reviewer 2 Report
In the manuscript titled: “Suppression of SHROOM1 Improves In Vitro and In Vivo Gene Integration by Promoting Homology-Directed Repair” by Zhao et al, the authors explore the role of SHROOM1in successful CRISPR-Cas9 knock-in gene editing. The authors discover that KD of SHROOM1 increases the efficacy of gene editing in an HDR dependent manner.
These data are very interesting since the function of SHROOM1 was not described in the context of gene editing.
Remarks:
- Figure 1d – where is SHROOM1 located in this analysis?
- For data from figure 1 provide an excel table with all the genes and the corresponding FC.
- Not clear why the first experiment was conducted in osteosarcoma cells and the second in HEK293T, please explain.
- Provide a supplementary excel table containing all the genes from your second experiment, including FC.
- I recommend to perform the siRNA experiments from figure 2, regarding SHROOM1 in a second cell line.
- Figure 3 – I recommend using two different siRNA against SHROOM1.
- Figure 4 – add the PCR bands for successful knock-in in the main figures of the manuscript.
- Figure 5 – please show the florescence images which show the intensity of knock-in, similar to figure 3c (for all 5 experimental groups).
Author Response
Dear Reviewer:
Thank you for your comments concerning our manuscript entitled “Suppression of SHROOM1 Improves In Vitro and In Vivo Gene Integration by Promoting Homology Directed Repair” (ID: ijms-862174). Those comments are all valuable and very helpful for revising and improving our paper, as well as the important guiding significance to our researches. We have studied comments carefully and have made correction which we hope meet with approval. Revised portion are marked in red in the paper. The main corrections in the paper and the responds to the comments are as flowing:
Responds to the reviewer’s comments:
Point 1: Figure 1d – where is SHROOM1 located in this analysis?
Thanks. The gene is marked in Figure 1d on page 3.
Point 2: For data from figure 1 provide an excel table with all the genes and the corresponding FC.
Thank for your notice. All genes and their corresponding FC are shown in the supplementary table S4 and noted in page 2, paragraph 4, line 15.
Point 3: Not clear why the first experiment was conducted in osteosarcoma cells and the second in HEK293T, please explain.
We have added following explanation to paragraph 1 of discussion on page 10. “Genome-wide screening in iDR-GFP U2OS by siRNAs exhibits high stability and accuracy. Many genes appear in our sight, such as CtIP and BRCA1, and may play important role in HDR maintaining activity. In order to exclude potential false positive and find potent HDR-suppressing genes, we use CRIPSR/Cpf1 and paired crRNAs to implement next screening. Considering the instability and lower HDR incident of BFP-reporter after transfecting CRISPR system in U2OS, we designed a dual-cut BFP reporter in HEK293T to carry on the subsequent screening. Finally, we found suppressing SHROOM1 could improve HDR ratio in dual-cut BFP 293T significantly.”
Point 4: Provide a supplementary excel table containing all the genes from your second experiment, including FC.
Thank for your notice, genes and their corresponding FC from the second experiment are shown in the supplementary table 5 and noted in page 4, paragraph 2, line 8.
Point 5: I recommend to perform the siRNA experiments from figure 2, regarding SHROOM1 in a second cell line.
Thanks a lot for your helpful suggestion. We used iDR-GFP U2OS to perform the siRNA experiment. After treated with siRNA and induced by Dox, the similar effect on HDR efficiency was generated in iDR-GFP U2OS. The related result is shown in supplementary Figure 3b and following description is added in page 4, paragraph 3, line 5. “Four siRNAs exhibited obviously increased HDR efficiency in dual-cut BFP 293T cells and iDR-GFP U2OS cells, especially SHROOM1.”
Point 6: Figure 3 – I recommend using two different siRNA against SHROOM1.
Thank for your suggestion. We have used additional siRNAs of SHROOM1 (named siSHROOM1’ or siSHROOM1-Homo-1576 shown in Table S3 ) or shroom1(named sishroom1’ or sishroom1-Mus-1394 shown in Table S3) to detect the HDR efficiency. As expected, similar increased HDR efficiencies were generated. The corresponding statistical diagram and fluorescence images are respectively shown in revised Figure 3, page 7 and supplementary Figure 4c, page 6. And the following description of the similar influence of HDR is added to Page 6, paragraph 1, line 14 in the main text. “The similar elevated HDR ratios were generated by another siRNA against SHROOM1 (siSHROOM1’) (Figure S4d)”.
Point 7: Figure 4 – add the PCR bands for successful knock-in in the main figures of the manuscript.
Thank for your notice. The PCR bands are added in Figure 4, while deleted in Figure S5. And corresponding legend are modified in page 8 of main text and page 5 of the supplementary document.
Point 8: Figure 5 – please show the florescence images which show the intensity of knock-in, similar to figure 3c (for all 5 experimental groups).
Thank for your suggestion, corresponding florescence images and flow charts are shown in Figure 5d on page 9.
Special thanks to you for your good comments.
